# Transcriptomic and Metabolomic Analysis of a Pseudomonas-Resistant versus a Susceptible Arabidopsis Accession

**DOI:** 10.3390/ijms232012087

**Published:** 2022-10-11

**Authors:** Isabel Orf, Hezi Tenenboim, Nooshin Omranian, Zoran Nikoloski, Alisdair R. Fernie, Jan Lisec, Yariv Brotman, Mariusz A. Bromke

**Affiliations:** 1Department of Life Sciences, Ben-Gurion University of the Negev, Beer-Sheva 8410501, Israel; 2Max Planck Institute of Molecular Plant Physiology, Am Mühlenberg 1, 14476 Potsdam, Germany; 3Bioinformatics Group, Institute of Biochemistry and Biology, University of Potsdam, Karl-Liebknecht-Str. 24-25, 14476 Potsdam, Germany; 4Department of Analytical Chemistry, Federal Institute for Materials Research and Testing, Richard-Willstätter-Straße 11, 12489 Berlin, Germany; 5Department of Biochemistry and Immunochemistry, Wroclaw Medical University, ul. Chałubińskiego 10, 50-367 Wrocław, Poland

**Keywords:** Arabidopsis, *Pseudomonas syringae* pv. *tomato*, resistance, systems approach

## Abstract

Accessions of one plant species may show significantly different levels of susceptibility to stresses. The *Arabidopsis thaliana* accessions Col-0 and C24 differ significantly in their resistance to the pathogen *Pseudomonas syringae* pv. *tomato* (Pst). To help unravel the underlying mechanisms contributing to this naturally occurring variance in resistance to Pst, we analyzed changes in transcripts and compounds from primary and secondary metabolism of Col-0 and C24 at different time points after infection with Pst. Our results show that the differences in the resistance of Col-0 and C24 mainly involve mechanisms of salicylic-acid-dependent systemic acquired resistance, while responses of jasmonic-acid-dependent mechanisms are shared between the two accessions. In addition, arginine metabolism and differential activity of the biosynthesis pathways of aliphatic glucosinolates and indole glucosinolates may also contribute to the resistance. Thus, this study highlights the difference in the defense response strategies utilized by different genotypes.

## 1. Introduction

The Gram-negative bacterium *Pseudomonas syringae* pv. *tomato* (Pst, Pseudomonas) infects a wide variety of economically important crops. Pseudomonas proliferates in the intercellular spaces of leaves of susceptible plants after entering through natural openings (e.g., stomata) or wounds. Infected leaves show necrotic patches [1]. The interaction between the model plant *Arabidopsis thaliana* (Arabidopsis) and Pseudomonas strains has been extensively studied [1]. Studies focusing on the Arabidopsis–Pseudomonas pathosystem have contributed to the elucidation of several molecular mechanisms playing a role in plant–pathogen interactions, such as the mechanisms of pathogen recognition in plants [2,3], the signal transduction pathways [4,5], and pathogen virulence and avirulence factors [6,7]. Lately, ‘-omics’ approaches were used to characterize the Arabidopsis–Pseudomonas pathosystem. Maleck et al. [8] investigated the transcriptional reprogramming during initiation of local defense and systemic acquired resistance (SAR) in Arabidopsis upon infection with Pseudomonas. Similarly, Tao et al. [9] focused on compatible and incompatible interactions. Further studies characterized transcriptomic changes related to pathogen-associated molecular pattern (PAMP) recognition [10], and type III secretion system effectors [11]. Microarray technology was used for analyzing gene expression during Arabidopsis immune responses upon the infection with Pseudomonas DC3000 [12]. Changes occurring in the global metabolite composition of *Arabidopsis* during the course of *Pseudomonas* infection were also reported. The main observation was an accumulation of phenolic and indolic compounds, as well as alterations in the abundance of amino acids, glucosinolates, and disaccharides [13]. In recent years, Lewis et al. studied transcriptional dynamics of Pst DC3000-induced immunosuppression in Arabidopsis Col-4 with high temporal resolution, which led to identification of motifs (ABRE, MycN, and G-box) enriched in genes rapidly suppressed in response to microbial-associated molecular pattern recognition [14]. Furthermore, they reported consistent overrepresentation of MYB-binding domains in promoter regions of differentially expressed genes(DEGs) [14]. The microarray dataset generated by Lewis et al. was used by an independent group in differential co-expression analysis [12]. Their Gene Ontology (GO) enrichment analysis revealed that the most significantly enriched GO term was “response to stimulus” as the most significantly enriched process, followed by “defense response” and “cell wall modification”. Additionally, they confirmed the importance of MYB51 in regulating the synthesis of camalexin in Pst-infected Arabidopsis [12,15].

Not every ecotype of Arabidopsis is equally susceptible to *Pseudomonas* infection. An analysis of the ability of *P. syringae* pv. *maculicola* to proliferate in 100 different Arabidopsis accessions showed that some accessions are up to 250-fold more resistant than the reference accession Columbia-0 (Col-0), whereas others are up to 2.5-fold more susceptible [16]. Natural variation in the perception of bacterial flagellin was assayed using 45 Arabidopsis accessions and showed great variance owed mostly to the abundance of protein components of MAMP perception pathways [17]. Natural diversity observed in the Arabidopsis immune response was used to identify crucial polymorphisms behind differential interaction of ZED1 with *Pseudomonas syringae* and *Xanthomonas campestris* effectors and its down-stream partners [18]. A broad genetic screening of ecotypes (including Col-0 and C24) for disease resistance-related nucleotide-binding leucine-rich repeat (NLR) proteins, which serve as intracellular immune receptors that recognize pathogen proteins, provided an interesting insight into natural resistance diversity. For instance, the highest sequence diversity in NLR genes was related with resistance to evolutionarily adapted biotrophic pathogens (e.g., *Hyaloperonospora arabidopsidis*) [19].

Mishina and Zeier (2007) have already demonstrated elevated resistance of C24 in comparison to Col-0 to infection with specific Pst DC3000 and a non-host strains of *P. syringae.* Their study on several Arabidopsis accessions suggested induction of salicylic acid accumulation and pathogenesis-related gene expression at inoculation sites, and that induction of these defenses is largely dependent on bacterial type III secretion system [20]. For our study, we chose a systems approach, using both transcriptomic as well as metabolomic analyses to study the Arabidopsis–Pseudomonas pathosystem in a comprehensive manner. Specifically, we characterized the difference between the susceptible Arabidopsis accession Col-0 and the resistant accession C24 in the reaction upon infection with *P. syringae* pv. *tomato* (Pst). Our study not only offers an integrated analysis of the defense response in Arabidopsis upon Pst infection on different systems levels, but also of the natural variation within this interaction. Our results show that these differences appear mainly on the level of salicylic-acid (SA)-dependent systemic acquired resistance (SAR), whereas responses of jasmonic-acid (JA)-dependent mechanisms are shared, albeit delayed in Col-0. Thus, our results provide some insights as to how differences in the reaction of Arabidopsis accessions may translate into resistance or susceptibility to Pst. 

## 2. Results

### 2.1. Appropriate Experimental Study to Dissect the Arabidopsis–Pseudomonas Pathosystem

Experimental designs for studying pathosystems have to take into account not only the effect of the studied pathogen, but also that of the inoculation method used. To distinguish the effects of the pathogen from the potential wounding responses that the widely used syringe-injection inoculation method may elicit, we first conducted an experiment in which we compared untreated plants, plants infiltrated with a mock solution, and Pst infiltrated plants. The reason for conducting this experiment was our concern that syringe infiltration might trigger wounding responses. In this experiment, we compared plant metabolite levels using gas-chromatography mass-spectrometry (GC-MS) at four time points (1, 3, 5, and 7 h post injection, Appendix A). We observed that for some metabolites such as succinic acid, maltose, phenylalanine, and the most important of the major defense hormones, salicylic acid (SA), the effect of mock injection was hardly distinguishable from that of Pst injection but was significantly different from the untreated plants. This was observed as early as 1 h post injection, and remained stable up to the last time point, 7 h post injection. Based on this experiment, we decided to use spray inoculation for our main experiment to avoid triggering defense responses in the plants that could affect/mask the response to Pst.

To obtain a comprehensive picture of Pst’s interaction with Col-0 and C24, we spray inoculated both accessions either with a MgCl_2_ solution (mock treatment) or with Pst suspension. Rosette tissue samples were taken from six independent replicates of both treatments at six time points: 2, 4, 8, 16, 24, and 48 h after the inoculation. This experimental setup allowed us to compare and contrast Col-0 and C24 with respect to their defense reactions against Pst. Samples were analyzed using the Affymetrix ST 1.0 microarray (Appendix A, GSE90852) for transcriptome analysis, and GC-MS (Appendix A) and liquid chromatography-MS (LC-MS, Appendix A) were used for the metabolome analysis (see Materials and Methods).

### 2.2. Significant Transcriptomic Differences in the Basal Gene Expression of SA-Dependent SAR in Col-0 and C24

We first focused on identifying large differences in time-resolved gene expression between Col-0 and C24 under control conditions. To find such differences between uninfected Col-0 and C24 plants, we averaged the log_2_-transformed gene expression levels of the six time points for each accession and subtracted the values of Col-0 from that of C24 to calculate log_2_-transformed ratios. Genes were considered to be differentially expressed (DEGs) when they showed a fold change > ±1.5 (log_2_ = 0.585) and a significance threshold of *p*-value < 0.01. With this method, we found 2366 genes to be differentially expressed between Col-0 and C24 (Appendix A, Sheet 1). Of those, 729 (around 30%) were more highly expressed in Col-0, whereas 1637 were more highly expressed in C24. GO-term enrichment analysis with the 2366 DEGs revealed enrichment for defense-related GO Biological Process (BP) terms (*p*-value < 0.05). All but 1of the top 20 GO BP terms were defense-related (Appendix A, Sheet 2), while 7of the top 20 GO BP terms were SA-related.

Inspecting the GO BP terms enriched in C24 DEGs, we found 17 of the top 20 terms to be defense-related, and again 7of the top 20 were SA-related (Table 1, full list in Appendix A, Sheet 3). The most significant enrichment in C24 DEGs was found for the term “GO:0009627 systemic acquired resistance”. Among the basal defense-related DEGs in C24 were many known role-players in basal defense and SAR: *PR1, FMO1, NIMIN1, PR5*, *WRKY* transcription factors (TFs), to name but a few [21,22,23,24]. SAR is known to be the result of predominately SA-mediated defense mechanism activation [25].

The most significant enrichment in Col-0 DEGs was found for the term “GO:0006952 defense response” (Table 2, full list in Appendix A, Sheet 3). Most of the significantly enriched GO BP terms were housekeeping processes such as “GO:0007165 signal transduction” or “GO:0007154 cell communication”. Interestingly, three of the top GO BP terms were related to the metabolism of the glucosinolate defense compounds.

### 2.3. The Basal Expression of SA-Related Genes in C24 under Axenic Conditions

The plants used in our transcriptomic analysis were grown in normal, non-axenic soil and growth chambers, and were therefore exposed to interactions with soil- and water-borne microorganisms. Although this exposure did not result in any visual signs of disease, it cannot be excluded that plant defense responses were triggered. Thus, part of our microarray results could reflect differences in the way Col-0 and C24 address the ambient response. It may even be claimed that C24’s higher basal resistance to Pst is in fact not basal at all, but rather a side effect of the response towards ambient interactions. To address this, we grew Col-0 and C24 for 14 days in sealed plates, under axenic conditions, and subjected them to microarray analysis using Affymetrix ST 1.0. The gene expression patterns of the axenic cultures experiment were in considerable agreement with the soil experiment. Altogether, 414 (17.2%) of the 2366 DEGs between Col-0 and C24 on soil showed opposite change in comparison to the plate experiment. When applying a threshold of the fold change > ±1.5 (log_2_ = 0.585), this percentage could be reduced to 3.2%. GO-term analysis of the 414 genes mentioned above did not result in any significant enrichment. Considering the different growth conditions in the two experiments (growth chamber and soil versus plate and sterile medium) and age of the sampled plants (45 days versus 14 days), the close similarity in expression values underscores the robustness and validity of our main experiment. The results also imply that any influence that ambient factors may have on the initial trigger of the defense response, before Pst infection, are minor and that the core DEGs between Col-0 and C24 are tightly regulated on the genomic level in different developmental stages and environments. For the comparison of the DEGs from Arabidopsis grown in soil and axenic cultures, see Appendix A.

### 2.4. The Aliphatic Glucosinolate Biosynthesis Pathway Is Differential between Col-0 and C24

Next, we compared the metabolite profiles of uninfected Col-0 and C24 plants using both GC-MS analysis for primary metabolites, as well as LC-MS analysis for secondary metabolites. As an example, metabolic changes at 2 h and 24 h post infection are provided in Figure 1 (see Appendix A for other timepoints). As early as 2 h after infection amino acids such as isoleucine, valine, serine, glycine, and threonine were accumulated in leaves of both ecotypes. Additionally, phenylalanine and leucine were on the higher levels in Col-0 only. On the other hand, C24 reduced leaf content of several other metabolites, including sucrose (Figure 1). Out of amino acids mentioned above, only serine and threonine were accumulated in the infected plants 24 h post infection in both ecotypes. Several metabolites such as galactinol, sucrose, pyruvic acid, and an unidentified analyte (Ukn_13), were over-accumulated in Col-0 and C24. The metabolic profile of infected Col-0 at this time point differs greatly from C24: 20 metabolites accumulated vs. 5 metabolites accumulated and 1 reduced in C24 (Figure 1).

Partial least squares discriminant analysis (PLS-DA) of primary metabolites in mock-treated plants showed a clear separation of the two accessions along dimension 1 (Figure 2). Ten analytes contribute most to the separation along the dimension 1: maleic acid, 1,6-anhydro-beta-D-glucose, erythritol, isopropyl-beta-D-thiogalactoside, ornithine, indole-3-acetonitrile, citrulline, or arginine, and two non-annotated compounds. We repeated the PLS-DA using secondary metabolites (Appendix A), and again found a clear separation of the two accessions along the dimension 1. A total of 4 of the 10 secondary metabolites contributing the most to the separation of the two accessions belong to the group of glucosinolates, specifically aliphatic glucosinolates.

Therefore, we closely inspected the levels of glucosinolates annotated in our secondary metabolite dataset. In total, we found 41 glucosinolates (Table 3). Most of the measured glucosinolates were more abundant in C24 compared to Col-0. Table 3 present statistically significant foldchange of glucosinolate concentration as well as these glucosinolates which were detected only in one accession 48 h post infection (more data in Appendix A). Out of 41 glucosinolates annotated in both studied accessions, 8 were not found in Col-0 (absent or below the detection level). These were 9-methylthiononyl glucosinolate, hydroxypropyl glucosinolate, glucobrassicanapin, 5-hexenyl glucosinolate, glucocappasalin, 5-(methylthio)pentyl glucosinolate, 4-phenylbutyl glucosinolate, and sinigrin (common names used when possible), whereas C24 plants did not containglucolepidiin. In leaves of Col-0, the greatest increase in content was observed for 6-heptenyl glucosinolate, followed by glucoiberverin (2.92- and 2.59-fold increase, respectively). In C24, changes of these glucosinolates were much smaller (28% and 26%, respectively) and below the significance threshold (Table 3). Most of the measured glucosinolates were more abundant in C24 compared to Col-0. For instance, 6-heptenyl glucosinolate and glucoiberverin in average were found in concentrations 540- and 34-fold higher in C24 than in Col-0. As suggested by the PLS-DA, an interesting pattern could be observed in aliphatic glucosinolates. While aliphatic glucosinolates of rather short chain length (propyl, butyl, pentyl) were more abundant in Col-0, longer chain lengths were found in C24 (hexyl, heptyl, octyl, nonyl) (Table 3). This pattern was found in both thio- as well as sulfinyl- side chains.

Previously, the ratio of long- to short-chain glucosinolateswas found to be characteristic of the “MAM phenotype” [26]. Indeed, the different glucosinolate chain lengths in Col-0 and C24 are reflected in the gene expression of methylthioalkylmalate (MAM) synthases 1 and 3 that catalyze the chain-elongation steps in the core glucosinolate biosynthesis pathway (Figure 3). Whereas no difference could be observed in the expression of MAM1 and all other genes of the pathway (BCAT4, CYP79F1, CYP83A1, SUR1, UGT74B1, SOT16, SOT17) between Col-0 and C24, MAM3 expression was much higher (four to eight-fold) in C24 compared to Col-0.

### 2.5. Col-0 and C24 Share JA-Dependent Responses upon Infection with Pst

We next analyzed the changes in the transcriptome and metabolome of Col-0 and C24 after infection with Pst. To obtain DEGs, we took the following approach: We first performed segmentation analysis (see Materials and Methods) and observed that the time series can be segmented into two intervals, each including three time points ((2, 4, 8 h), (16, 24, 48 h)) (Appendix A). Within each interval, the time points were pooled, i.e., they were considered as replicates. In addition, we called a gene differentially expressed (DE) at time point I if the respective log-fold change of expression in time point i and i + 1 was not within the range of [mean ± 2 × standard deviation] of the log-fold changes over all genes.

Using this method, the total number of DEGs upon infection at each time point was comparable between Col-0 and C24 (approximately 1700 DEGs at each time point; Figure 4A). The number of DEGs shared between both accessions varied between 348 and 498 at 2 h and 48 h post-inoculation, respectively (see central diagonal in Figure 4A). We further focused on defense-related DEGs among these lists. Among the defense-related shared DEGs at each time point between the two accessions (central diagonal in Figure 4B, Appendix A), we found many genes of ethylene and JA biosynthesis. Approximately 50% of the common defense-related DEGs at late time points of the infection (from 16 h onwards) were genes involved in JA-mediated processes (central diagonal in Figure 4C). Within the genes significantly changed at the later time points in both accessions were allene oxidase cyclases and synthase 1 through 3 (*AOC1-3*); [27,28], the jasmonate-zimdomain proteins (*JAZ1*, *JAZ5*, *JAZ7*, *JAZ8*, *JAZ10*), jasmonic acid carboxyl methyltransferase (*JMT*) [29], and the 12-oxophytodienoate reductase 3 (*OPR3*) [30,31].

### 2.6. JA-Related Responses after Infection with Pst Are Delayed in Col-0

Some of the JA-related genes caught our attention because of their delayed response in Col-0 in comparison to C24. To find more genes with delayed response, we filtered the dataset for those genes that were significantly changed in early time points of the infection in C24 (2 h, 4 h, 8 h, defined as Interval 1) and late time points of the infection in Col-0 (16 h, 24 h, 48 h, defined as Interval 2) (Table 4). Within this list, two groups/clusters of gene expression can be found. Group I consists of 13 genes that are only significantly changed in Interval 1 in C24, group II consists of 31 genes significantly changed in both intervals in C24. Among all 44 genes, 17 genes were related to defense processes (bold in Table 4), of which five were transcription factors and therefore regulators of defense responses (*MYB13*, *MYC2*, *RAP2.6*, *Rap2.6L*, *WRKY75*). 

The key compound of jasmonate signaling is jasmonoyl-(L)-isoleucine (JA-Ile), which is formed in a specific conjugation reaction of JA with isoleucine by the enzyme jasmonate-amido synthetase (*JAR1*; *AT2G46370*) [32,33]. The expression of JAR1, as well as expression of jasmonate-dependent transcription regulator coronatine insensitive 1 (*COI1*, *AT2G39940*) did not change in the studied Arabidopsis accessions upon Pst infection. We observed an accession-specific timing in the regulation of JA-Ile catabolism genes: jasmonoyl-isoleucine-12-hydroxylase CYP94B3 (*AT3G48520*) was over-expressed in Col-0 at 24 h and 48 h; hydrolase *IAR3* (*AT1G51760*) was upregulated in C24 from the earliest time point and in Col-0 from eight hours after the infection; hydrolase *ILL6* (*AT1G44350*) was induced in both Pst-sprayed accessions in the second interval (Appendix A).

We measured the levels of JA at 8 h, as the last time point before the strong induction of JA-mediated processes, and 16 h, as the first time point of said induction. The levels of JA confirmed both the comparable behavior of JA-related genes in Col-0 and C24 upon Pst infection, as well as the delay in the JA-mediated response in Col-0 (Figure 5). JA levels tended to be higher in infected samples, showing the comparable behavior in Col-0 and C24, although the difference to the mock treatment was not significant (Figure 5). No observable difference in JA was found between the 8 h and 16 h time point of C24. JA in infected Col-0 at 8 h was lower than in C24 at 8 h, reflecting the delay in the response in Col-0, although the difference was only marginally significant (*t*-test, *p* = 0.049). Salicylic acid content in Pst infected Col-0 leaves was significantly higher than in the mock control. However, this increase represents only a small fraction of the SA content in C24, where SA did not change between treatments (Figure 5).

### 2.7. Pst Induces SA-Dependent SAR-Related Gene Expression in Col-0

Among the accession-specific DEGs at each time point (according to GO BP), we found biphasic behavior in Col-0 with peaks at 8 h and 48 h post-infection that were absent in C24 (Appendix A). Interestingly, of the 244 defense-related DEGs in Col-0 at 48 h post-infection, 95 were related to SAR (Figure 4B, Appendix A). In contrast, only 10 of the 77 defense-related DEGs in C24 48 h post-infection were related to SAR (Figure 4B, Appendix A). The lack of SAR-related genes in the late defense response of C24 is probably due to their high basal expression (124 DEGs related to SAR were found in C24 in the comparison of uninfected plants in contrast to 7 DEGs related to SAR in Col-0). We therefore concluded that SAR-mediated processes may be not inducible in C24, or their inducibility is at least reduced. This becomes even more evident considering the behavior of SAR-related genes in Col-0 and C24 over the time course (mock-treated and infected plants) (Figure 6). The expression of many well-known SAR-related genes such as *CERK1*, encoding for a LysM receptor kinase essential for chitin elicitor signaling in Arabidopsis [34], was generally higher in C24 than in Col-0 and did not change upon infection. The expression of SAR-related genes in Col-0, on the other hand, was low in mock-treated samples and early time points of the infection and increased at late time points of the infection, often reaching expression levels similar to C24 at 48 h post-infection. Likely, Col-0 by increasing expression upon infection compensates for its lack of basal expression of defense-related genes. This applies to the above-mentioned *CERK1*, but also to a flavin-dependent monooxygenase 1 (*FMO1*) shown to be an essential component of SAR [23]; an extracellular beta-1,3-glucanase (*BGL2*) [35]; a transcriptional coregulatory activating SA-dependent defense gene (*NPR1*) [36]; a paralog of NPR1 and SA receptor (*NPR3*) [37]; pathogenesis-related 1 that is activated by SA in basal defense (*PR1*) [24]; another member of the pathogenesis-related protein family (*PR5*) [21]; an E3 ubiquitin ligase involved in the regulation of ubiquitination during the defense response to microbial pathogens (*RING1*) [38]; and transcription factors *WRKY38*, *WRKY46*, *WRKY54*, and *WRKY70* [39,40,41]. These genes could be classified according to the pattern of observed changes into five classes: (A) genes that show higher expression in C24 than in Col-0, but do not significantly change in C24, but do in Col-0; (B) genes that show higher expression in C24 than in Col-0, show some significant changes in C24, but mostly in Col-0; (C) genes that show higher expression in C24 than in Col-0, change significantly at early time points in C24, and in late time points in Col-0; (D) genes that show no significant changes, but the same trend as genes of category A; (E) genes that show higher expression in C24 than in Col-0, change significantly at all time points in C24, and in most time points in Col-0 (Figure 6A–E).

Interestingly, the findings related to SAR genes in Col-0 were only partially reflected by the metabolite levels of SA, pipecolic acid (Pip), and glycerol-3-phosphate (G3P), all known to be involved in SAR (Figure 5 and Figure 7) [42,43]. As expected, based on gene expression, SA levels were much higher in C24 compared to Col-0 and did not change upon infection (Figure 5). Conversely, in Col-0, SA increased significantly in Pst-infected plants. Levels of Pip, too, were higher in C24 in comparison to Col-0, but increased significantly at late time points in infected samples of both accessions (Figure 7). G3P showed similar levels in Col-0 and C24 up to the 8 h time point (Figure 7). In late time points, however, basal levels of G3P were higher in C24 than in Col-0. Whereas G3P did not increase upon infection in C24, a significant increase could be observed in infected Col-0 at 48 h, reaching the levels of G3P in C24.

Pip and its derivatives *N*-hydroxy-Pip and *N*-hydroxy-Pip-O-glycoside are small molecules important to pathogen defense, in particular for the establishment of SAR [44,45]. There is a significant difference in the expression pattern of genes involved in biosynthesis of these compounds between Col-0 and C24. Pst-infected C24 maintained 2–3 higher expression of *ALD1*, *FMO1*, and *UGT76B1* throughout the time course of the experiment, whereas in Col-0 the expression of these genes was induced much later, at 24 h and 48 h. One should note the much higher normalized basal expression level of the genes in C24 (Appendix A).

There was no effect of Pst infection on the regulation of the G3P synthesis; the expression of *GLY1* (DHAP dehydrogenase; *AT2G40690*) and *GLI1* (glycerol kinase; *AT1G80460*) did not change. As demonstrated by Yu et al. [46], the G3P-mediated SAR is likely dependent on *AZI1* and *DIR1*, and G3P might act upstream of these two proteins. *AZI1* expression in Col-0 and C24 shows different patterns. In Pst-infected Col-0.Expression of *AZI1* gene increased at 8 h and 16 h, after which it returned to the mock level. In C24, expression of the gene was repressed in infected plants at later stages (24 h and 48 h). The expression of *DIR1* in both ecotypes was similar and did not differ significantly between mock- and Pst-treated plants (Appendix A). Increased expression of a G3P-consuming G3P dehydrogenase 2 (*PGDH2, AT1G17745*; Appendix A) in Col-0 in only was observed.

## 3. Discussion

Here, we report on the dynamic response of the metabolome and transcriptome of resistant (C24) and susceptible (Col-0) Arabidopsis accessions in response to Pst infection. Although our results show major reprograming of the plant metabolome and transcriptome of the plants upon Pst infection, they strongly indicate that basal differences in gene expression and levels of specific metabolites explain a great part of the observed differences between the accessions.

### 3.1. SAR-Related Differences between Col-0 and C24 Play a Major Role in Determining Resistance to Pst

Many SA-related defense genes were found to be more highly expressed in C24 compared to Col-0 under non-infected conditions (Appendix A), in agreement with previous studies [47]. It is known that SA plays an important role in SAR, a mechanism conferring long-lasting protection against a broad spectrum of pathogens [48]. Active SAR renders plants into a primed state in which they more quickly and effectively activate defense responses [49]. Therefore, SAR is considered to function mostly in the defense against secondary infections [50]. Here, “GO:0009627 systemic acquired resistance” was found to be the most significantly enriched GO term of basal C24 DEGs (Table 1). We not only observed that transcripts of SAR-related genes were already highly expressed in uninfected C24 plants, but also that their expression remained stable after infection, exhibiting no additional induction of SAR in C24. This could not only be seen on transcript level, but also in the levels of G3P and SA measured in the plants (Figure 4 and Figure 6B). We therefore conclude that SAR is permanently active in C24. Moreover, SAR in C24 seems to be not inducible and C24 may even be unable to deactivate this pathway, although the latter statement needs to be studied in more detail in the future. Col-0 expresses only low levels of SAR-related genes when uninfected. Upon infection with Pst, these genes are strongly induced (Figure 6, Appendix A). This behavior is not only in agreement with the previously reported reaction of Col-0 to Pst [4,42,51], but can also be interpreted as a compensation for the initial lack of SAR activity. As in C24, the SAR signature in Col-0 can be seen not only at the transcript but also at the metabolite level, e.g., SA and G3P (Figure 5 and Figure 7). Constitutively active SAR may give C24 a considerable lead over Col-0 in repelling Pst intrusion and spreading within the leaves. C24 exhibits higher resistance compared to Col-0 not only against Pst, but also against other, non-bacterial pathogens such as cucumber mosaic virus and the fungus downy mildew (*Hyaloperonospora arabidopsidis*) [52,53], underscoring the broad spectrum of its basal defense.

### 3.2. Nitrogen Metabolism and Aliphatic Glucosinolates

Our results show that SAR is not the only defense-related difference between Col-0 and C24. In our analysis of primary metabolites, we found the differences between Col-0 and C24 explained to a great extent by metabolites belonging to the urea cycle and arginine biosynthesis pathway. C24 showed higher levels of citrulline/arginine, as well as ornithine, in comparison to Col-0, but lower levels of their precursor fumaric acid (Appendix A). Arginine can serve as nitrogen storage and as a precursor of NO and polyamines, both playing a role in defense mechanisms [54]. The expression of arginase 2 (*AT4G08870*), an enzyme which degrades arginine to urea and ornithine, was strongly induced by the Pst infection in Col-0 only. The induction was observed in Interval 2. The base non-induced expression level of *AT4G08870* was higher in Col-0 than in C24, in which the expression upon challenge with Pst did not change at all (Appendix A). Polyamine biosynthesis in Arabidopsis begins with arginine, which is converted by arginine decarboxylase to agmatine [55]. Arginine decarboxylase 2 (*ADC2, AT4G34710*), was induced by the Pst infection and the effect was stronger and appeared earlier in Col-0 than in C24 (Appendix A). Plant arginine decarboxylase 2 is an important element of the defense system, as *adc2* knock-out mutants were shown to be more susceptible to a Pst DC3000 infection [56]. It is worth mentioning that in the *adc2* mutant putrescine content was reduced, but with addition of exogenous putrescine. the pathogen resistance was rescued [56]. In our experiments, more than 2-fold increase in putrescine content in leaves was observed as early as 24 h after the infection (Appendix A), which is in line with previous studies on the role of polyamines in Arabidopsis-Pst pathosystem [57]. In addition, arginine is suggested to be used to fine-tune development and defense mechanisms against stress [55]. If and how the differences in arginine metabolism in Col-0 and C24 translate into the higher resistance of C24 requires further analysis.

Camalexin is a sulfur-containing tryptophan-derived secondary metabolite and is considered to be the major phytoalexin involved in biotic responses in *A. thaliana.* Camalexin is a phytoalexin found in Arabidopsis, where it deters bacterial and fungal pathogens [58]. We measured the levels of indole-3-acetonitril (IAN), which is an intermediate in the biosynthesis pathway of camalexin. It starts with oxidation of tryptophan by two functionally redundant cytochrome P450 enzymes, CYP79B3 and CYP76B2 (encoded by *AT2G22330* and *AT4G39950*, respectively). These genes showed a different regulation pattern between the ecotypes leading to increased expression in Interval 2. In the following enzymatic step controlled by another cytochrome P450 enzyme, CYP71A13 (*AT2G30770*) indole-3-acetaldoxime is converted into an unstable intermediate, which further is conjugated with glutathione by the glutathione-S-transferase (GSTF6) to synthesize GS-IAN. The gene in C24 was also expressed on higher level in Pst than in mock treated plants throughout the time course. On the other hand, in Col-0, the *CYP71A13* expression pattern did not differ between Pst and mock in time points 2 h through 24 h. At 48 h post-infection, the expression remained elevated in Pst-treated Col-0, whereas in mock plants, it returned to the base level. After the conjugation step, GS-IAN is hydrolyzed to Cys-IAN in a process in catalyzed by γ-glutamyl peptidases GTP1 and GTP3 [59]. Gene expression for both peptidases was not affected by the Pst infection. Finally, the last two reactions of the biosynthesis pathway leading to camalexin are catalyzed by PAD3/CYP71B15 enzyme. The basal expression of PAD3 gene is higher in C24 than in Col-0 and elevated expression was observed in both ecotypes sprayed with Pst except for the last (48 h) time point in C24, where the difference did not pass the significance threshold. Cheval et al. [60] observed similar increase of PAD3 transcript abundance in leaves of Pst-sprayed Arabidopsis accession Columbia, though in their experiment the expression returned to the base level within 24 h. Nevertheless, camalexin content following the Pst infection increased by about 20-fold and 60-fold at 24 h post infection and 48 h post infection, respectively [60].

According to a model proposed by Müller et al. [61], indole-3-acetonitrile (IAN) can be synthesized by myrosinase or by an unknown protein through degradation of indolyl-3-methyl-glucosinolate. Next, IAN can be converted to precursors of camalexin or indole-3-carboxylic acid by CYP71B6. Expression of CYP71B6 gene did not change during the infection (Appendix A). Interestingly, C24 contained less IAN than Col-0 during the experiment. As the result of the Pst spraying, a significant increase in IAN above the mock level was observed in Col-0 at 16 h and 48 h, whereas in C24, after initial significant drop (4 h), the IAN level in infected plants rose above the mock within 48 h. Thus, the camalexin biosynthesis pathway seems not to contribute substantially to the higher resistance of C24 against the Pst infection.

Among the differences in secondary metabolites between Col-0 and C24, those found in the group of aliphatic glucosinolates were the most striking. Glucosinolates are produced from amino acids by cycles of chain elongation [62]. The levels of aliphatic glucosinolates in Col-0 and C24 differed largely in respect to their chain length. Col-0 exhibited higher levels of short-chain aliphatic glucosinolates, while in C24 long-chain glucosinolates were more abundant (Table 3). Upon infection, levels of long-chain aliphatic glucosinolates remained stable in C24 (Appendix A). In Col-0, on the other hand, the levels of long-chain aliphatic glucosinolates increased at late time points of the Pst infection. The difference in chain length in aliphatic glucosinolates between uninfected Col-0 and C24 could be explained by the expression of MAM3 (Figure 3). It was previously shown that MAM3 catalyzes chain elongation in both short- and long-chain aliphatic glucosinolates, whereas MAM1 catalyzes fewer cycles of chain elongation, resulting in short-chain aliphatic glucosinolates [63]. Interestingly, although the levels of long-chain aliphatic glucosinolates increased, MAM3’s expression did not change significantly in infected Col-0 plants compared to control plants. Thus, the potential factor triggering this increase might not be active at the level of gene expression, but at a different molecular level such as, e.g., posttranscriptional modification.

While all glucosinolates are potentially reactive defense compounds, it was shown previously that aliphatic glucosinolates of higher chain length exhibit higher toxicity against the pathogenic ascomycete *Sclerotinia sclerotiorum* [64]. A second study showed that decreasing the side-chain length of aliphatic glucosinolates as well as the extent of hydroxylation increased feeding by the beetle *Psylliodes chrysocephala* [65]. Our observations—higher basal levels of long-chain aliphatic glucosinolates in C24 and increase in their levels in Col-0 upon infection—underscore the putative role of these compounds in resistance against Pst.

Furthermore, the similarity in the behavior observed for components of SAR and the levels of long-chain aliphatic glucosinolates in Col-0 and C24 hints towards an interaction between the two pathways, the nature of which has been debated in the past: while exogenous application of SA was shown to massively induce biosynthesis of aliphatic glucosinolates [66], Mewis et al. [67] showed that constitutively blocking SA-signaling using an *npr1* mutant increased glucosinolates in Col-0. The proposed interaction between SAR and the biosynthesis of long-chain aliphatic glucosinolates therefore remains unresolved and requires further in-depth analysis.

In our study, levels of branched-chain amino acids (BCAAs), isoleucine and valine, as well as of threonine(the precursor of isoleucine) increased in infected plants of both accessions at early time points (2 h to 4 h; Appendix A). This seems to be a part of defense-mechanisms as isoleucine is probably, though this is not unequivocally confirmed, the precursor of isoleucic acid (ILA, 2-hydroxy-3-methyl-valeric acid), a molecule which plays a role in modulating defense-response, as demonstrated through exogenous application of ILA in experiments of von Saint Paul et al. [68]. Infection with either virulent or avirulent strains of Pst triggered a decrease of ILA abundance in Col-0 plants. Interestingly, related compound leucic acid (LA) was not affected by the Pst infection [69]. Maksym et al. [69] observed no significant correlation between BCAA and ILA or LA. There are two proposed defense-enhancing modes of action of ILA: SA-dependent and SA-independent. In first, ILA competitively inhibits SA glucosylation of UGT76B1. Thus, the ILA-dependent enhanced defense response and pathogen resistance can be attributed to the suppression of the attenuation of SA glucosylation. In the former, ILA engages superoxide formation and root growth inhibition [70].

Among the metabolites changing in response to infection, the non-protein amino acid Pip particularly caught our attention. Increase in Pip at late time points of the infection could be seen in both accessions (Figure 6), but it was more predominant in C24. Interestingly, metabolite content of SA, Pip, and G3P (Figure 4 and Figure 6), all known to be involved in SAR, only partially reflected findings related to the expression of SAR genes in Col-0.

We observe no changes in expression of GLY1, GLI1,and DIR1 genes upon Pst infection, which is in line with the results of Yu et al. [46]. There is a difference between the studied accessions in terms of G3P content. Pst-infected Col-0 had a tendency to accumulate more G3P than mock-treated plants. Yet, these differences were not significant, except for the last time point, when it passed the significance threshold. Moreover, *PHDH2*, encoding for an enzyme which utilizes G3P, was induced only in infected Col-0 as early as 4 h post infection, and the increased expression lasted until the 48 h time point. No significant changes were observed in C24. Similar results, more than six-fold increase in expression of PHDH2 gene, were obtained from Col-0 sprayed with Pst DC3000 by Yang et al. [71]. Both the accumulation of G3P and increased expression of PGDH2 in Col-0 might suggest activation of serine supply pathway in this accession upon infection. PGDH2 is a serine-regulation insensitive enzyme which starts the pathway of G3P metabolism into serine. Thus, G3P is required for ammonium and sulfur assimilation as well as tryptophan biosynthesis [72,73]. Expression of other isoforms of the enzyme was not affected by the treatment (Appendix A).

Following the increase in ALD1 expression, expression of isochorismate synthase 1 (*ICS1*), *FMO1*, and *NPR1* increases, leading ultimately to rising SA levels and the full establishment of SAR not only locally, but throughout the plant. While the accumulation of SA at the infection site seems to be independent of Pip, it is required for the activation of *ICS1* and thus SA biosynthesis in distal tissue. Our observations for the behavior of the elements of the SAR feedback loop for signal amplification after infection in Col-0 correspond to the literature, although increases in *NPR1* expression were not significant (Figure 5 and Figure 6). In *A. thaliana*, ~10% of defense-related SA is produced by the cytosolic phenylalanine ammonia lyase pathway, whereas the main pathway responsible for approximately 90% of SA is derived from isochorismate. This pathway starts with the plastid-localized isochorismate synthase 1 (ICS1). The study by Rekhter et al. suggests that following step EDS5 (Enhanced Disease Susceptibility 5, *AT4G39030*) exports isochorismate from the plastid into the cytosol, where PBS3 (*avrPphB Susceptible 3, AT5G13320)* metabolizes it to isochorismate-9-glutamate, which spontaneously decomposes or is enzymaticaly cleaved into SA [74,75]. In our study, *PBS3* expression in Col-0 is more than two-fold higher eight hours after spraying with Pst suspension (Appendix A), which is in agreement with measurements of SA, as at this time point its content in Col-0 nearly doubled in Pst-challenged plants (Figure 4). Moreover, *PBS3* seems to also be feedback regulated by SA [76]. Before the activation of *ICS1* expression in Col-0 and at the same time as *PBS3*, two pathogen-induced transcription factors, SAR deficient 1 (*SARD1*) and CaM-Binding Protein 60-like G (*CBP60g*) are activated. SARD1 and CBP60g are known to physically bind to an *ICS1* promoter and to regulate the expression of *ICS1* and are required for pathogen induction of SA synthesis [77]. A similar pattern of expression in Col-0 can be seen for *enhanced disease susceptibility 1 (EDS1)* and its sequence-related interaction partner phytoalexin deficient 4 (*PAD4*). EDS1 and PAD4 proteins are key regulators of Arabidopsis in basal immunity and each has a conserved esterase/lipase catalytic triad embedded within an α/β-fold hydrolase topology [78]. EDS1 might be involved in indirect 9-oxo nonanoic acid and azelaic acid accumulation in SAR [79]. In summary, whereas in Col-0 expression of discussed here genes of SA synthesis and related regulation increased with the time, in Pst-infected C24, the genes expression generally followed the pattern observed for the mock-sprayed plants.

Whereas transcription of ALD1 was induced at all time points of the infection in C24, a late elevation of Pip levels in C24 after infection was observed. This could be explained by its ongoing metabolism towards *N*-hydroxy-Pip. Defense gene induction by Pip is dependent on functional FMO1 (expression was elevated in C24 as early as 2 h post infection), suggesting the requirement of *N*-hydroxy-Pip formation and accumulation upon pathogen attack for SAR-related gene expression [44,80]. Moreover, it was demonstrated that local application of *N*-hydroxy-Pip was able to induce its biosynthesis pathway expression, genes encoding for ALD1, SAR-deficient 4 (SARD4; ornithine cyclodeaminase), and FMO1, both locally and systemically [44]. From this, a conclusion could be drawn that long-distance movement of *N*-hydroxy-Pip from infiltrated sites could regulate the synthesis of Pip and *N*-hydroxy-Pip via a positive feedback loop, just like Pip. Further, it has been suggested that similarly to SA glucosylation and storage of SA-Glc in response to pathogen infection, O-glucosylation of *N*-hydroxy-Pip could serve as a stabilization mechanism and that its glucoside may be the primary storage form of the molecule in Arabidopsis [44]. Recently published parallel studies [45,81] demonstrated the key role of *N*-hydroxy-Pip-*O*-β-glucosylating activity of a uridine diphosphate-dependent glucosyltransferase (UGT76B1; *AT3G11340*) for development of SAR in Pst-infected Arabidopsis. The significance of UGT76B1 for SAR is supported by the fact that this enzyme can glucosylate SA as well changing its availability as a regulatory compound [82]. In our experiment, Col-0 and C24 exhibited different expression patterns of the *UGT76B1* gene: early response and constantly elevated expression in infected C24, slow response and two-step increase in infected Col-0. Differential expression patterns of *ALD1*, *FMO1,* and *UTG76B1* might suggest existence of different co-regulation mechanisms in studied Arabidopsis accessions. Higher levels of metabolites (Pip, SA) in C24 as well as elevated baseline expression of Pip metabolism genes suggest existence of a defensive preparedness in C24 in contrast to Col-0. Experiments of Bechtold et al. suggest that this alerted state (increased consumption of lysine and phenylalanine precursors for synthesis of Pip and SA) imposes a penalty on rosette biomass of C24 plants, while the reproductive fitness remains similar to Col-1 [47].

### 3.3. JA-Related Responses against Pst Are Shared between Accessions, though Are Delayed in Col-0

*MYB13* was shown to be expressed in a JA/ethylene-dependent manner in Arabidopsis infected with *Botrytis cinerea* [83,84]. MYC2 was reported to modulate JA-dependent functions in Arabidopsis [85], for example, the expression of *PDF1*.2 that showed the same expression pattern as *MYC2* in our experiment (Table 4) [86,87]. *RAP2.6* and *Rap2.6L* were shown to be responsive to JA, SA, abscisic acid, and ethylene in addition to salt and drought [88,89,90]. *WRKY75*, though suggested to play a role in plant defense mechanisms, was shown to be mainly a modulator of phosphate acquisition [91]. Because of the striking representation of transcription factors modulating JA-dependent reactions, we concluded that some of the JA-related responses after Pst infection are delayed in Col-0 compared to C24.

JA-related defense responses to Pst at the level of gene expression and the levels of JA itself were shared between Col-0 and C24 (Appendix A, Figure 5). At late time points of the infection (from 16 h onwards), more than half of the shared defense-related DEGs in Col-0 and C24 were JA-related (Figure 4B,C, Appendix A). JA-mediated processes therefore seem to play an important role in the sustained defense response of both Col-0 and C24. JA is a fatty-acid-derived plant hormone involved in processes that are known to protect plants against insect herbivores; JA is therefore considered to play an important role in the response to plant wounding. However, tomato plants deficient in JA showed increased susceptibility to *Pseudomonas* infection [92].

Pst was shown to produce a phytotoxin, coronatine, that is believed to mimic 12-oxo-phytodienoic acid (OPDA), a precursor of JA [93]. The virulence property of coronatine is largely attributed to its ability to mimic the function of bioactive jasmonate, JA-Ile [94]. The *coi1* mutants, which lack the key integrator of jasmonate signaling, exhibited increased resistance to Pst and higher levels of SA upon infection [95]. However, several Arabidopsis mutants unable to activate JA-mediated responses showed normal susceptibility towards Pst [96]. It was shown that resistance to Pst in the *coi1* mutants is correlated with a strong activation of PR1 expression and accumulation of SA after infection. Moreover, restriction of Pst growth in the *coi1* mutants was fully dependent on SA, indicating that SA-mediated processes are required for Pst resistance [96]. Therefore, the increased resistance towards Pst in *coi1* mutants that were generated in a Col-0 background is in good accordance with our results. Both *coi1* and C24 show higher resistance to Pst due to higher activity of SA-mediated defense pathways. SA-signaling also has an antagonistic effect on JA-signaling [97]. Thus, high concentration of SA in C24, may also explain the elevated expression of JA-Ile catabolism genes in C24 from the very beginning of our experiment, as well as lack of changes in expression of JA-Ile biosynthesis genes. One should note that in SA-deficient *NahG* plants infected by Pst, JA accumulates to 25-fold higher levels than control, leading to induction of JA-responsive genes [97]. These findings suggest that, in response to a pathogen that can induce synthesis of both SA and JA, cross-talk is used by the plant to adjust the response in favor of the more effective pathway. Our results suggest that C24 infected with Pst favors the SA-driven response, whereas Col-0 activates both, but off-set in time. The high overlap of JA-related DEGs at later time points suggests that Col-0 is able to catch up with C24 in regard to JA responses (Figure 4, Appendix A).

## 4. Materials and Methods

### 4.1. Plant Material and Growth Conditions

*Arabidopsis thaliana* plants of two accessions, Col-0 and C24, were grown in soil (potting compost) in individual pots under a 16/8 h light/dark regime at 16 °C. Two days prior to bacterial inoculation, plants were transferred to a different growth chamber and were grown under a 14/10 h light/dark regime at 23 °C until harvesting. Light intensity in all chambers was 150 µE m^−2^ s^−1^.

Axenic cultures of *Arabidopsis* were prepared by filling square Petri dishes with agar ½ MS medium [98]. Sterilized seeds (washed with EtOH and sodium hypochlorite) were then planted on the medium. Plants in a growth chamber were grown with 12 h day (22 °C, relative humidity 60%) and 12 h night (20 °C, relative humidity 75%) at 250 µE/m^2^ for 14 days.

### 4.2. Bacterial Growth

*Pseudomonas syringae* pv. *tomato* strain DC3000 bacteria were grown on modified King’s B medium: 10 mg/mL peptone, 1.5 mg/mL K_2_HPO_4_, 15 mg/mL glycerol, 0.6% (*v*/*v*) 1 M MgSO_4_ 7H_2_O) on plates at 28 °C for two days, then transferred to liquid King’s B medium for one night [99]. All media were supplemented with 100 µg/mL rifamycin. Overnight cultures were washed with 10 mM MgCl_2_, then serially diluted in this buffer.

### 4.3. Inoculation and Sampling

For inoculation by infiltration, bacteria were diluted in 10 mM MgCl2 to 5 × 10^8^ cfu mL^−1^, then supplemented with 0.02% (*v*/*v*) Silwet and infiltrated into selected leaves using a needleless syringe. Mock solution was identical, save for bacteria. Plant material was harvested at 1, 3, 5, and 7 h post-infection. Untreated plants served as control. Samples were harvested from individual plants and immediately frozen in liquid nitrogen before being powdered using a cryogenic grinding robot (Labman Automation, Stokesley, UK).

For spray inoculation, bacteria were diluted in 10 mM MgCl_2_ to 5 × 10^8^ cfu mL^−1^, then supplemented with 0.02% (*v*/*v*) Silwet and sprayed evenly to saturation upon the plants. Mock solution was identical, save for bacteria. Plant material was harvested at 2, 4, 8, 16, 24, and 48 h post-infection. Samples were harvested from individual plants and immediately frozen in liquid nitrogen before being powdered. Each time point and condition was studied with six biological replicates.

### 4.4. Transcriptomics

RNA was extracted from frozen plant material using the RNeasy kit (Qiagen, Hilden, Germany). RNA was hybridized to Affymetrix Gene Arabidopsis 1.0 ST microarray chips (Affymetrix, Santa Clara, CA, USA). Hybridization and detection was performed by ATLAS Biolabs (Berlin, Germany). Microarray data are deposited in the NCBI Gene Expression Omnibus (GEO) repository (http://www.ncbi.nlm.nih.gov/geo, accessed on 26 September 2022) under GEO accession GSE90852.

Data preprocessing included normalization of raw data using the RMA algorithm [100,101]. Multivariate time-series (MTS) data can be used to determine the time segments corresponding to critical biochemical events which are reflected in the coupling of the system’s components [102,103]. A segmentation of MTS data is the partition of the time-series into a series of consecutive segments. Here we applied a regression-based formalization of the segmenting MTS data based on temporal changes in covariance structure. In this approach, for each time point, the explanatory power of the precedent time points is estimated from the fused LASSO regression. The breakpoints are then defined as time points for which the explanatory power of the precedent time points isnegligible; likewise, breakpoints have small explanatory contribution for their following sequence of time points.

Given a single replicated measurement for gene expression time series data, we combined the results of two sets of differential analysis for the purpose of stringent analysis and to avoid loss of information. Segmentation analysis allows deciphering intervals of time in which the behavior of most of the genes shows stable trends. A time series can then be divided into segments by finding time point(s) at which the behavior of most of the genes is drastically altered. We then pooled the data from the time points which fall in an interval, which we in turn used to find differentially expressed genes (DEGs). To ensure the reliability of pooling time points as replicates within an interval, the log-fold changes between each time point in the treatment versus control were obtained for each gene. Genes with large coefficient of variation (CV > 0.5) of the log-fold changes within an interval were filtered out. The R package *limma* was then applied to detect the DEGs at each interval with respect to the corresponding interval in control data set [104]. The same analysis was performed to identify DEGs between two intervals in treatment in comparison to the control. The log-fold changes at each time point were calculated for each gene between the treatment and the control data set. At each time point, the genes with log-fold changes within the range of [mean ± 2 × standard deviation] were filtered out. If a gene was identified as DE in two consecutive time points (i and i + 1), it was considered as DEG at time point i.

GO enrichment analysis was performed with help of the R package *GOstats* [105] and allowed us to find the biological processes and molecular functions which are over-enriched in the clusters determined from the clustering analysis. The cut-off value for significance level was considered as 0.05 after FDR correction. We defineddefense-related genes as genes that belong to GO BP terms containing “defense”, “response to other organism”, “response to biotic stimulus”, “bacterium”, “systemic”, “jasmonic”, “ethylene”, and “salicylic”.

### 4.5. Metabolomics

Gas chromatography-mass spectrometry (GC-MS) analysis was performed as described previously [106]. Metabolite levels were determined in a targeted fashion using the R environment and the *TargetSearch* 1.28.1 package [107]. Metabolites were selected by comparing their retention indices (±2 s) and spectra (similarity 85%) against the compounds stored in the Golm Metabolome Database [108]. Each metabolite is represented by the observed ion intensity of a selected unique ion which allows for a relative quantification between groups. Metabolite data were log_10_-transformed to improve normality [109] and normalized to show identical median peak sizes per sample group. Significance of changes in metabolite levels between Pst-infected and mock-treated plants was assessed using Student’s *t*-test.

Ultra-performance liquid chromatography coupled with high resolution mass spectrometry (UPLC-HRMS) was performed on a C8 reverse-phase column coupled to an Exactive mass spectrometer (Thermo Scientific, Bremen, Germany) in positive and negative ionization mode according to the method of Hummel et al. [110]. Processing of chromatograms, peak detection, and integration were performed using REFINER MSH 5.3 (GeneData, Basel, Switzerland). Processing of mass spectrometry data included the removal of the fragmentation information, isotopic peaks, as well as chemical noise. Obtained features (*m/z* at a certain retention time) were queried against an in-house database for further annotation. Remaining unknown features were cross-checked with online databases KNApSAcK v1.200.03 and PubChem using an in-house-developed database search tool (Golm Biochemical Space, GOBIOSPACE, http://gmd.mpimp-golm.mpg.de/webservices/wsGoBioSpace.asmx, accessed on 29 September 2021) for putative annotation.

For hormone measurement, 250 mg of powdered rosette leaves of *A. thaliana* was used for extraction. The extraction was performed at 4 °C for 30 min in 6 mL 2% (*v*/*v*) formic acid in water. The supernatant was separated from the plant debris by a 5 min centrifugation at 3000× *g*. The procedure was repeated once. The pooled supernatant was subjected to reverse-phase (C18) solid phase extraction (StrataX 33 µm particles in a 30 mg bed, Phenomenex, Aschaffenburg, Germany). SPE columns were washed with 2 mL methanol and equilibrated using 2 mL of 2% (*v*/*v*) formic acid in water. After loading of the samples, residual salts were washed off with 2% formic acid (*v*/*v*). Phytohormones were eluted using methanol and the eluate was dried using a speed vacuum concentrator. The dried pellet was re-suspended in 50% MeOH:H_2_O (*v*/*v*). MS-based analysis was performed on a triple-quadrupole mass spectrometer (ABI 3000, Applied Biosystems, Foster City, CA, USA) connected to an ACQUITY UPLC (Waters, Milford, CT, USA) equipped with a BEH C18 reverse-phase column (100 × 2.1 mm i.d., 1.8-µm particle size; Waters, Milford, CT, USA), operated at a flow rate of 400 µL min^−1^. Solvent A consisted of 0.1% formic acid in water and solvent B consisted of 0.1% formic acid in methanol. The gradient started at 62% A followed by a 7-min linear gradient to 10% A. The column was re-equilibrated for 3 min at 62% A for 3 min. The eluate was continuously monitored in negative ion mode using MRM (multiple reaction monitoring). Each compound, jasmonic acid (JA) and salicylic acid (SA) was identified and quantified based on two fragments (SA 137 *m*/*z* to 93 *m*/*z* and 85 *m*/*z*; JA 209 *m*/*z* to 15 *m*/*z* and 59 *m*/*z*). The parameters of the mass spectrometer for the analysis were set to −4.5 kV electro spray voltage at a temperature of 350 °C. The dwell time was set to 50 ms and pause between mass ranges was set to 5 ms. The collision energy was determined for each compound using authentic reference compounds, which were obtained from OlChem (Olomouc, Czech Republic).

Following the recommendations regarding best reporting practices on annotation, quantification in mass spectrometry-based metabolomics published in Alseekh et al. (2021) [111], information required for transparency in measurement and metabolite annotation and documentation are made available in Appendix A.

### 4.6. Data Analysis

Statistical analyses and graphical representations (Student’s *t*-test, ANOVA, PLS-DA), boxplots, heatmaps) were performed using the R software environment (http://cran.r-project.org/) and Microsoft Excel 2010. Partial least squares discriminant analysis (PLS-DA) between genotypes was performed using *mixOmics* 6.0.0 package.

## 5. Conclusions

Using a multi-omics approach, we were able to dissect differences between the two Arabidopsis accessions Col-0 and C24 with regard to their resistance toPst—before and after infection. We identified the constant upregulation of components of SAR in C24 as the main factor for its higher resistance towards Pst, although other factors such as nitrogen metabolism and aliphatic glucosinolates may contribute to its defense-primed state. Except for the SAR-related metabolite Pip, SAR mediated responses seemed to be not further inducible upon infection in C24. Upon infection, we found JA-mediated responses to be delayed in Col-0 when compared to C24. Even though Col-0 seemed to be able to catch-up with JA-mediated responses at later time points, the delay could have a great impact on spreading of Pst in the leaves and therefore on the severity of the infection. Our integrated metabolomics and transcriptomics analysis also revealed that indolic glucosinolates might be part of the JA-mediated defense pathway. Col-0’s lack of priming at early time points of the infection and especially the time point of the inoculation might therefore be causal for the increased susceptibility of Col-0 to Pst infection.

## Figures and Tables

**Figure 1 ijms-23-12087-f001:**
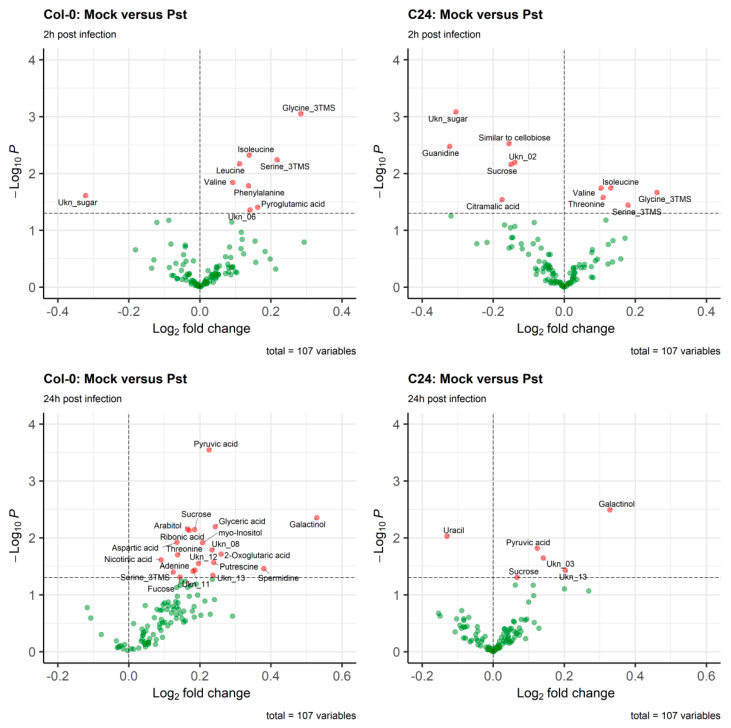
Volcano plot of the primary metabolism metabolites (107 analytes) from Col-0 and C24 2 h and 24 h after the infection with Pst. The plot displays values of log_2_-transformed ratio of metabolite levels(Pst: mock) against the negative log_10_ of the *t*-test’s *p*-value. The horizontal dashed line represents *p*-value = 0.05. “Ukn”: unidentified metabolite.

**Figure 2 ijms-23-12087-f002:**
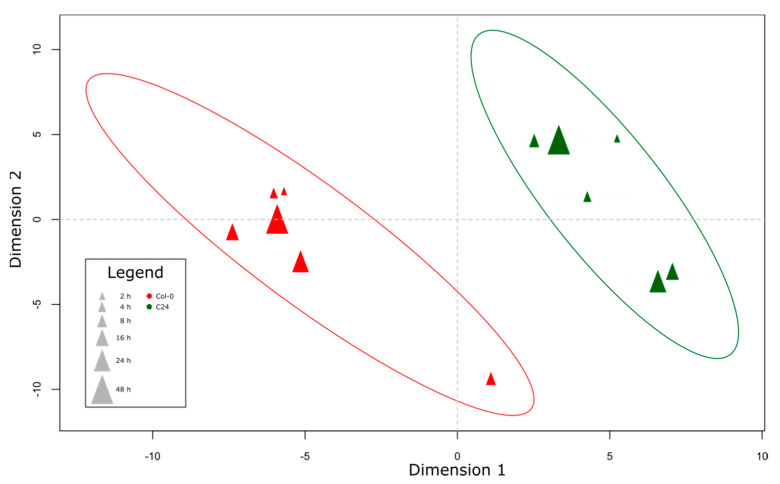
Score plot of PLS-DA of Col-0 and C24 mock treated plants based on primary metabolite profiles. Samples in the plot represent averaged metabolite profiles of 6 independent replicates.

**Figure 3 ijms-23-12087-f003:**
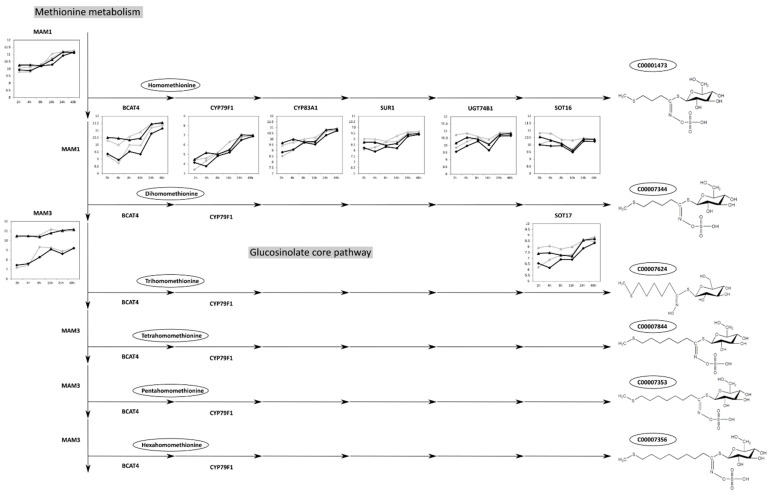
Expression of genes found in the methionine-derived glucosinolate core pathway in mock treated and Pst-infected Col-0 and C24 plants. Gene expression at time points 2 h, 4 h, 8 h, 16 h, 24 h, and 48 h marked as triangles (C24) or diamonds (Col-0) connected with gray (mock treatment) or black (Pst-infection) line. Abbreviations: *MAM1* = *AT5G23010, BCAT4* = *AT3G19710, CYP79F1* = *AT1G16410, CYP83A1* = *AT4G13770, SUR1* = *AT2G20610, UGT74B1* = *AT1G24100, SOT16* = *AT1G74100, SOT17* = *AT1G18590, MAM3* = *AT5G23020.* Pathway structure according to KEGG (http://www.genome.jp/kegg/, accessed on 29 September 2021). Structures of glucosinolates taken from KNApSAcK (http://www.knapsackfamily.com/KNApSAcK/, accessed on 29 September 2021).

**Figure 4 ijms-23-12087-f004:**
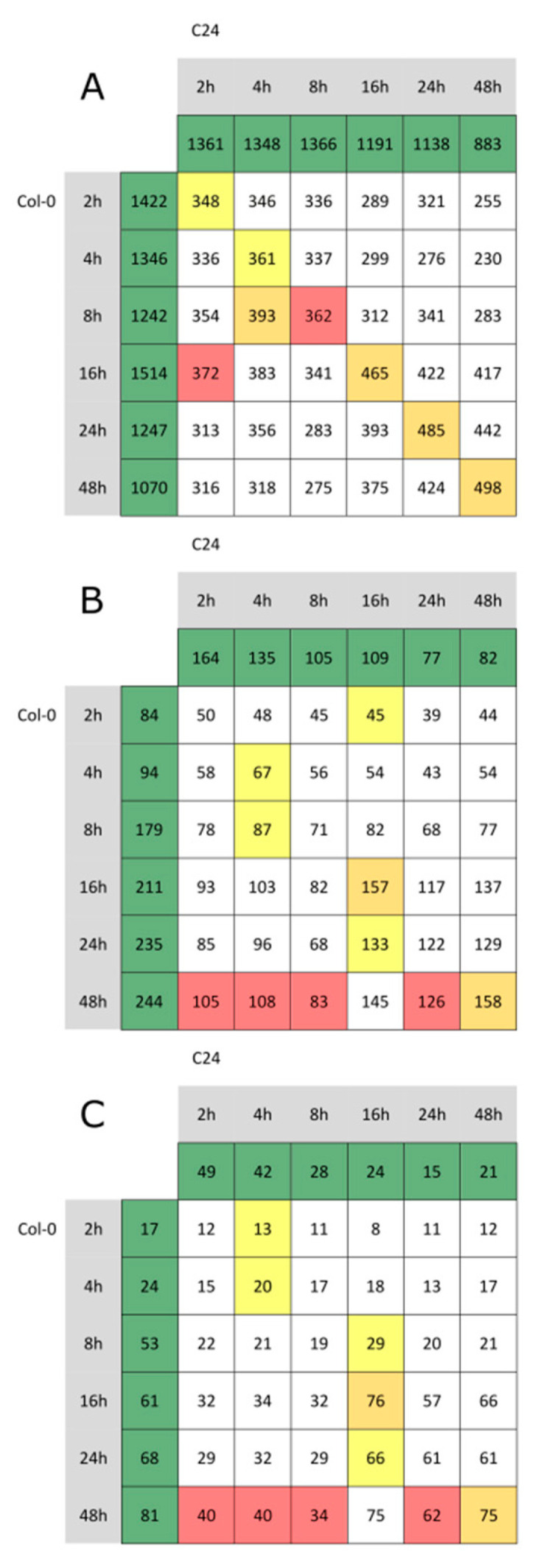
Numbers of differentially expressed genes (DEGs) in Col-0 and C24 after Pst infection.In green boxes, number of unique DEGs for each ecotype and time point. Numbers at the crossing of columns and rows represent DEGs shared by both accessions. Colors in the matrix: yellow, highest number in each row; red, highest number in each column; orange, highest number in both row and column. (**A**) Differentially expressed genes at each time point. (**B**) Defense-related DEGs at each time point. (**C**) JA-related DEGs at each time point.

**Figure 5 ijms-23-12087-f005:**
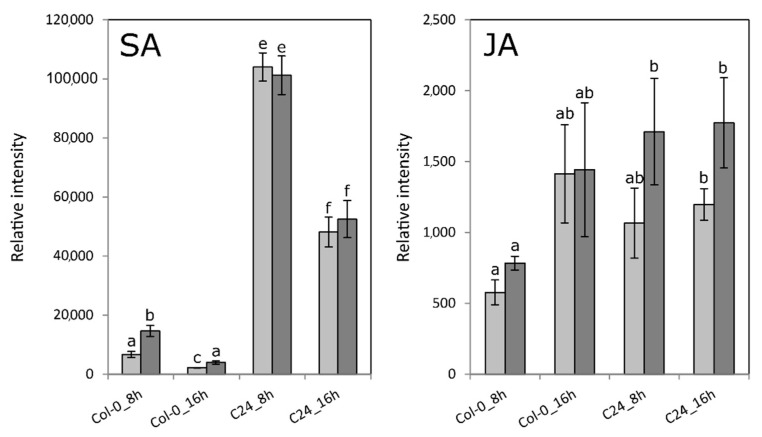
Levels of salicylic acid (SA) and jasmonic acid (JA) in Pst-infected and mock-treated Col-0 and C24 samples. Depicted are average intensities of four independent replicates. Error bars represent standard error. Significant differences between samples of Pst infection and mock treatment (*t*-test, *p* < 0.05) are indicated by different letters. Colors: light gray bars (mock treatment), dark gray (Pst infection).

**Figure 6 ijms-23-12087-f006:**
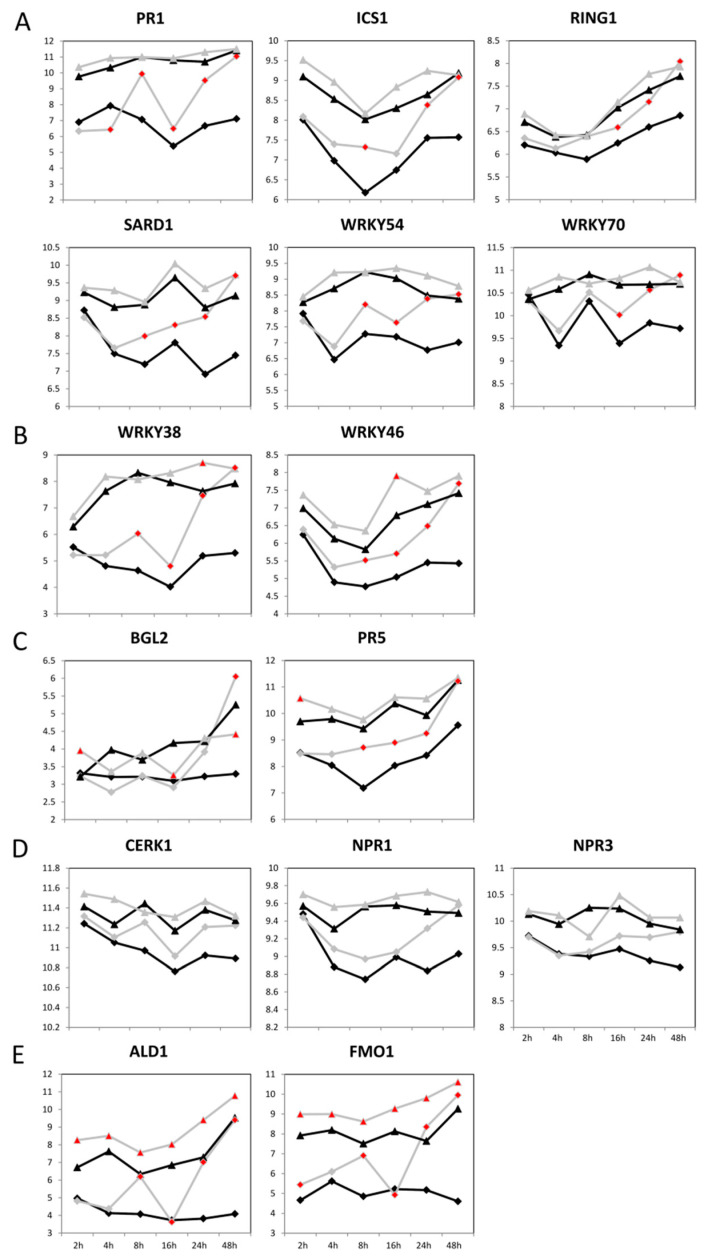
Gene expression of systemic acquired resistance (SAR)-related genes in mock-treated and Pst-infected Col-0 and C24 plants. Gene expression levels at given time points (2 h, 4 h, 8 h, 16 h, 24 h, and 48 h) are marked as triangles (C24) or diamonds (Col-0) and connected with gray (mock treatment) or black (Pst infection) line. Red filled symbols mark significant difference between mock-treated and Pst-infected plants. Panes: (**A**) Genes that show higher expression in C24 than in Col-0, do not significantly change in C24, but in Col-0. (**B**) Genes that show higher expression in C24 than in Col-0, show some significant changes in C24, but mostly in Col-0. (**C**) Genes that show higher expression in C24 than in Col-0, change significantly at early time points in C24, and in late time points in Col-0. (**D**) Genes that show no significant changes but the same trend as genes of category A. (**E**) Genes that show higher expression in C24 than in Col-0, change significantly at all time points in C24, and in most time Abbreviations: *ALD1* = *AT2G13810, BGL2* = *AT3G57260, CERK1* = *AT3G21630, FMO1* = *AT1G19250, ICS1* = *AT1G74710, NPR1* = *AT1G64280, NPR3* = *AT5G45110, PR1* = *AT2G14610, PR5* = *AT1G75040, RING1* = *AT5G10380, SARD1* = *AT1G73805, WRKY38* = *AT5G22570, WRKY46* = *AT2G46400, WRKY54* = *AT2G40750, WRKY70* = *AT3G56400*.

**Figure 7 ijms-23-12087-f007:**
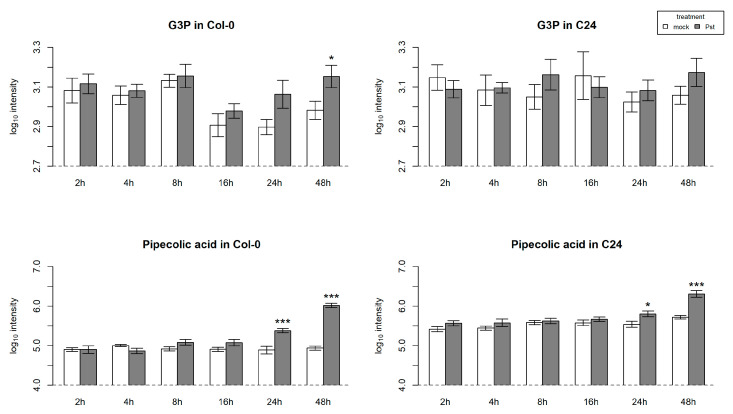
Levels of glycerol-3-phosphate (G3P) and pipecolic acid in Pst-infected and mock-treated Col-0 and C24 plants. Depicted are average values of log_10_ transformed peak intensities (5–7 independent replicates) with error bars of standard error. Significant differences between samples of Pst infection and mock treatment are indicated by asterisks (*t*-test; *p* < 0.05 “*”, *p* < 0.005 “***”).

**Table 1 ijms-23-12087-t001:** Significantly enriched gene ontology biological process (GO BP) terms in basal gene expression in C24: Enrichment analysis was performed using GOstats. The cut-off value for significance level was considered as 0.05 after FDR correction. Listed are the 20 most significantly enriched GO BP terms. Count = number of genes significantly changed in the group of genes belonging to the respective GO term, size = number of genes belonging to the respective GO term. Gray background highlights GO BP terms related to defense responses.

	GO BP ID	*p*-Value	Count	Size	Term
1	GO:0009627	6.35 × 10^−53^	128	442	systemic acquired resistance
2	GO:0009697	5.88 × 10^−49^	85	207	salicylic acid biosynthetic process
3	GO:0009814	1.16 × 10^−47^	134	532	defense response, incompatible interaction
4	GO:0009696	1.90 × 10^−47^	86	220	salicylic acid metabolic process
5	GO:0042537	8.94 × 10^−45^	90	258	benzene-containing compound metabolic process
6	GO:0006952	4.69 × 10^−44^	249	1644	defense response
7	GO:0002376	1.78 × 10^−42^	179	977	immune system process
8	GO:0045087	1.90 × 10^−42^	166	859	innate immune response
9	GO:0006955	4.89 × 10^−42^	166	865	immune response
10	GO:0071446	6.40 × 10^−35^	93	353	cellular response to salicylic acid stimulus
11	GO:0009751	1.12 × 10^−34^	108	469	response to salicylic acid stimulus
12	GO:0009607	1.15 × 10^−34^	208	1413	response to biotic stimulus
13	GO:0009863	2.31 × 10^−34^	92	351	salicylic acid mediated signaling pathway
14	GO:0051707	3.09 × 10^−34^	207	1412	response to other organism
15	GO:0031348	6.87 × 10^−33^	79	273	negative regulation of defense response
16	GO:0031347	1.27 × 10^−31^	112	539	regulation of defense response
17	GO:0080134	1.07 × 10^−30^	113	560	regulation of response to stress
18	GO:0009862	6.30 × 10^−30^	72	250	systemic acquired resistance, salicylic acid mediated signaling pathway
19	GO:0051704	2.42 × 10^−29^	231	1802	multi-organism process
20	GO:0050832	4.46 × 10^−29^	84	344	defense response to fungus

**Table 2 ijms-23-12087-t002:** Significantly enriched gene ontology biological process (GOBP) terms in basal gene expression in Col-0:Enrichment analysis was performed usingGOstats. The cut-off value for significance level was set at 0.05 after FDR correction. Count = number of genes significantly changed in the group of genes belonging to the respective GO term, size = number of genes belonging to the respective GO term. Gray background highlights GO BP terms related to defense responses.

	GO BP ID	*p*-Value	Count	Size	Term
1	GO:0006952	2.79 × 10^−4^	63	1644	defense response
2	GO:0007165	4.80 × 10^−4^	71	1948	signal transduction
3	GO:0044700	8.90 × 10^−4^	72	2027	single organism signaling
4	GO:0023052	9.02 × 10^−4^	72	2028	signaling
5	GO:0007154	4.15 × 10^−3^	78	2367	cell communication
6	GO:0016145	4.35 × 10^−3^	3	14	S-glycoside catabolic process
7	GO:0080028	5.70 × 10^−3^	2	5	nitrile biosynthetic process
8	GO:0006591	8.41 × 10^−3^	2	6	ornithine metabolic process
9	GO:0015824	9.54 × 10^−3^	6	74	proline transport
10	GO:0015804	1.15 × 10^−2^	6	77	neutral amino acid transport
11	GO:0032890	1.16 × 10^−2^	2	7	regulation of organic acid transport
12	GO:0050898	1.16 × 10^−2^	2	7	nitrile metabolic process
13	GO:0051952	1.16 × 10^−2^	2	7	regulation of amine transport
14	GO:0070417	1.22 × 10^−2^	3	20	cellular response to cold
15	GO:0000304	1.52 × 10^−2^	2	8	response to singlet oxygen
16	GO:0019674	1.52 × 10^−2^	2	8	NAD metabolic process

**Table 3 ijms-23-12087-t003:** Glucosinolate content change in leaves of Arabidopsis Col-0 and C24 within 48 h after treatment. In case of several glucosinolates, an alternative (isobaric) annotation is provided. Asterisks next to values represent *t*-test statistics *p*-values; * <0.05, ** <0.01, *** <0.005 Abbreviations: n.d., not determined.

	Content Fold Change after 48 h	
KNApSAcK ID: Name(s)	Col-0	C24
	Mock vs. Pst	Mock vs. Pst
C00007850: 5-Oxooctyl glucosinolate; Glucocappasalin	n.d.	1.63
C00007855: 4-Phenylbutyl glucosinolate	n.d.	1.53
C00007801: 5-Hexenyl glucosinolate	n.d.	1.37
3-Hydroxy-5-(methylthio)pentyl glucosinolate	n.d.	1.21
C00001463: Glucobrassicanapin; 4-Pentenyl glucosinolate	n.d.	1.19
C00007340: 3-Hydroxypropyl glucosinolate	n.d.	1.16
C00007817: 2-Hydroxypropyl glucosinolate		
C00007826: 1-Methyl-2-hydroxyetyl glucosinolate; Glucosisymbrin		
C00001488: Sinigrin; 2-Phenylethyl glucosinolate;	n.d.	1.11
Allyl glucosinolate		
C00007845: 9-Methylthiononyl glucosinolate	n.d.	1.01
C00007800: 6-Heptenyl glucosinolate	2.92 **	1.28
C00001473: Glucoiberverin;	2.59 ***	1.26
3-(Methylthio)propyl glucosinolate		
C00000125: Glucobrassicin; Glucobrassicine; 3-Indolylmethylglucosinolate	1.71 ***	2.17 *
C00007856: 3-Phenylpropyl glucosinolate	1.67 **	1.26
C00000130: 5-Hydroxyglucobrassicin; 5-Hydroxy-3-indolylmethylglucosinolate	1.63	2.03 *
C00007348: 6-Methylsulfinylhexyl glucosinolate; Glucohesperin	1.53 *	1.24
C00007813: 3-Hydroxy-6-(methylthio)hexyl glucosinolate		
C00007352: 7-(Methylsulfinyl)heptyl glucosinolate; Glucoibarin	1.52 *	1.29
C00007624: 5-Methylthiopentyldesulfoglucosinolate	1.47 *	1.21
C00007355: 8-Methylsulfinyloctyl glucosinolate; Glucohirsutin	1.41 *	1.13
C00007593: 5-Methylthiopentylglucosinolate; Glucoberteroin	1.32 *	1.22
C00007353: 7-Methylthioheptyl glucosinolate	1.31 *	1.19
C00007356: 8-Methylthio-octyl glucosinolate	1.28 *	1.07
C00001474: Glucolepidiin; Ethyl glucosinolate	1.16	n.d.

**Table 4 ijms-23-12087-t004:** Delayed transcriptional changes after Pst infection in Col-0 compared to C24: Displayed are log_2_-transformed expression values of Pst-infected samples after subtraction of expression in mock-treated samples. Gray background = significantly different from mock treatment in this interval. Group I = significantly changed only in interval 1 in C24, Group II = significantly changed in both intervals in C24.

Gene Annotation	Group	2 h Col-0 Pst—Mock	4 h Col-0 Pst—Mock	8 h Col-0 Pst—Mock	16 h Col-0 Pst—Mock	24 h Col-0 Pst—Mock	48 h Col-0 Pst—Mock	2 h C24 Pst—Mock	4 h C24 Pst—Mock	8 h C24 Pst—Mock	16 h C24 Pst—Mock	24 h C24 Pst—Mock	48 h C24 Pst—Mock
		Interval 1			Interval 2			Interval 1			Interval 2		
*ABO3*	I	0.103	−0.012	0.366	−0.181	1.630	1.265	0.971	0.393	0.764	0.071	0.589	0.536
*ACT11*	I	−0.226	−0.402	−0.731	−0.359	−0.850	−1.324	−0.627	−0.724	−0.560	−0.053	−0.024	−0.773
*AOC3*	II	0.328	0.529	0.892	0.897	1.573	2.488	0.967	0.540	0.347	1.386	1.594	1.313
*ASB1*	I	0.298	−0.187	0.006	0.799	1.482	1.588	0.966	0.865	0.963	−0.168	1.132	−0.674
*AT1G26390*	II	0.685	0.642	1.478	0.340	0.852	3.186	1.371	0.819	0.858	0.868	0.676	2.555
*AT1G53885*	II	0.449	0.307	0.441	2.016	2.292	3.491	0.609	0.437	1.076	2.354	2.846	3.425
*AT1G53885*	II	0.449	0.307	0.441	2.016	2.292	3.491	0.609	0.437	1.076	2.354	2.846	3.425
*AT1G60730*	II	0.321	0.378	0.520	0.452	0.524	1.035	0.684	0.747	0.406	0.690	0.510	0.788
*AT1G66310*	I	−0.296	−0.121	−0.794	−0.718	−1.015	−0.734	−0.016	−0.885	−0.933	0.352	−0.472	0.443
*AT2G45220*	II	0.376	0.369	2.129	0.467	0.965	3.166	0.987	1.035	1.201	0.818	0.546	1.904
*AT3G04000*	II	0.529	0.432	0.664	0.359	0.487	1.505	0.574	0.721	0.735	0.686	1.116	1.529
*AT3G11340*	II	0.311	1.576	2.301	1.246	3.784	4.448	1.297	1.275	1.021	1.130	2.061	0.305
*AT3G22600*	II	0.324	0.524	2.631	1.069	1.626	1.798	1.040	0.859	0.477	0.387	0.883	1.069
*AT3G46080*	I	0.082	−0.065	0.103	−0.019	1.822	2.888	1.302	0.915	1.227	0.472	0.883	1.278
*AT4G03130*	I	0.383	−0.324	−0.452	−0.726	−0.209	−1.041	−0.513	−0.414	−1.063	0.567	0.210	0.324
*AT4G11890*	I	0.251	0.398	1.576	0.830	1.309	1.781	0.466	0.915	0.396	0.404	0.188	0.098
*AT4G19970*	II	−0.096	−0.208	1.366	0.529	0.393	1.702	0.977	0.687	1.105	0.142	0.742	1.559
*AT5G05600*	II	0.111	0.545	0.909	1.480	1.805	3.007	0.852	0.593	1.127	1.921	1.887	1.873
*AT5G08240*	II	0.644	0.272	0.675	0.497	0.655	1.840	0.949	0.698	0.572	0.485	0.566	0.800
*AT5G39050*	I	0.168	0.613	0.472	0.551	0.919	1.364	0.808	0.598	0.581	0.255	0.576	0.674
*AT5G57510*	II	0.984	0.034	0.666	0.159	0.914	3.287	1.357	0.702	0.672	0.654	1.405	2.171
*CEN2*	II	−0.228	0.843	1.387	0.228	2.256	2.978	0.562	1.072	0.688	0.771	1.000	1.479
*COBL5*	I	0.023	0.641	0.809	0.533	0.778	1.461	0.567	0.681	0.553	0.448	0.525	0.302
*DUR*	II	0.534	0.791	1.592	0.092	1.595	3.018	0.785	0.803	0.729	0.730	0.979	1.989
*EDS1*	I	−0.275	−0.142	0.000	0.429	0.690	1.035	1.260	0.528	0.346	0.326	−0.263	−0.433
*FMO1*	II	0.779	0.480	2.049	−0.297	3.178	5.345	1.072	0.803	1.119	1.146	2.160	1.341
*GRX480*	II	−0.088	0.007	1.703	1.759	2.910	3.277	0.987	0.613	0.316	0.846	1.046	0.796
*GSTU11*	II	0.029	0.257	0.254	0.333	1.101	3.677	1.264	0.997	0.128	2.348	2.694	2.566
*GSTU4*	II	0.999	0.931	1.966	1.391	1.722	4.467	1.929	1.435	1.548	1.520	1.509	0.904
*GSTU8*	I	−0.586	−0.769	1.098	0.937	1.166	1.823	1.408	1.644	1.064	1.143	0.616	0.585
*IAR3*	II	−0.053	0.397	0.932	0.884	1.251	2.349	0.860	0.533	0.611	0.727	0.986	1.138
*KCO2*	I	−0.318	0.669	−1.301	−1.217	−0.333	−0.673	−0.033	−1.172	−0.582	−0.079	−0.793	−0.726
*LHT7*	II	−0.610	0.736	2.174	1.331	3.433	3.421	0.126	1.278	0.406	1.260	1.436	0.584
*MYB13*	II	0.059	0.560	0.600	0.328	0.797	1.428	0.569	0.875	0.743	0.386	0.958	0.935
*MYC2*	II	−0.030	0.155	0.367	1.191	1.256	1.525	0.791	0.782	0.190	1.131	0.931	0.983
*PDF1.2*	II	−0.351	0.428	2.226	2.240	1.757	2.317	1.654	1.651	2.028	1.620	1.277	0.082
*RAP2.6*	II	−0.201	−0.194	0.100	0.302	1.627	3.460	1.092	0.340	0.895	0.679	0.745	1.586
*Rap2.6L*	II	0.595	0.889	0.897	0.431	1.205	2.564	0.747	1.115	0.810	0.579	0.975	1.987
*RLP43*	I	0.503	0.117	1.137	0.388	1.122	1.656	0.829	0.937	0.422	0.084	0.557	0.447
*SBT3.3*	II	−0.498	0.952	1.016	0.524	0.655	2.469	1.394	0.949	1.221	1.041	1.584	1.582
*TPS04*	II	0.253	0.916	0.453	1.528	2.856	5.219	1.063	1.163	0.861	2.564	1.961	1.249
*UGT73D1*	II	0.123	−0.107	0.044	0.770	0.723	2.176	0.469	1.061	0.903	0.544	1.313	0.924
*WAK3*	II	0.359	0.272	1.733	1.003	1.688	2.445	0.824	0.669	0.489	0.276	0.767	0.962
*WRKY75*	II	0.759	0.181	1.547	0.269	1.214	4.190	1.208	1.271	0.733	0.886	0.674	1.676

## Data Availability

Data presented in this article will be available on reasonable request.

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
