# Peer review of "Transcriptomic and Metabolomic Analysis of a Pseudomonas-Resistant versus a Susceptible Arabidopsis Accession"

_ijms, 2022, doi:10.3390/ijms232012087_

Round 1
Reviewer 1 Report
Review to: Transcriptomic and Metabolomic Analysis of a Pseudomonas-Resistant versus a Susceptible Arabidopsis Accession
The aim of the work was to highlight differences in the defense response strategies in two Arabidopsis genotypes. Background of their study was that Col-0 is known to be more susceptible to infection with Pst and C24 seems to be more resistant. The authors nicely compared dynamic responses in the metabolome and transcriptome of C24 and Col-0 in response to Pst infection. They clearly show that the basal level of gene expression is a major part of the higher resistance of C24 and seems to prime the plant to activate SAR more quickly and effective. The JA-related responses are shared in both lines but are somehow delayed in Col-0. In addition, arginine metabolism, and differential activity of the biosynthesis pathways of aliphatic glucosinolates and indole glucosinolates may also contribute to the resistance.
The topic of the work is of current interest and the experimental approach is suited to unravel so far unknown aspects of Pst resistance in C24. As previous work already indicated that the basal level of priming in C24 might be of special relevance, the authors conducted control experiments with axenically grown plants, to avoid any contact with microorganisms that could evoke priming. In addition, different inoculation methods were tested, strengthening the work.
I have a few questions and comments that might improve the work further:
In line 705f it is said that each time point and condition was studied with six biological replicates. Is this true for all data shown in any figure or table? I recommend adding the information on the number of replicates to each figure and table legend.
Table 1 and table 2: For a better understanding the GO BPs of C24 and Col-0 should be edited and processes or genes that occur twice should be shortened, for example Table 2, items 6,7,and 8 are identical except for the description, this could be shortened. Table 1, item 8 (innate immune response) is a subgroup of item 9 (Immune response), only show one.
It in not fully clear from the table descriptions how the values were calculated. Was it Pst vs mock treatment at timepoint zero? Please add this information
Section 2.3: Are there any datasets available? Maybe it was overlooked to add these to the manuscript or supplements, at least I could not find a link to any results.
Figure 1B: The names are difficult to read, maybe without "unknown" metabolites or another arrangement of the letters these would be better accessible. What is the meaning of “107 variables” ?
Table 3: Showed a very long list which not easy to read. Maybe formatting was lost.
Figure 3: Here in many cases the mock samples follow the Pst treated samples and expression increases. This is not clear to me, please explain. Figure legend: “Gene expression of genes”, remove the first “Gene”.
Table 4: Please mention why there are some genes written in bold letters.
The discussion is very long and detailed, shortening would make it easier accessible to the reader.
It would be nice to have a picture of the infected vs mock treated plants. Are there symptoms visible after 48h ?
Data handling: Are the omics data available to the public, except for the supplemental exel files?
Author Response
Dear Reviewer,
Thank you for your detailed review of the manuscript. Following your comments and suggestions we have applied listed below changes to our manuscript.
1) In line 705f it is said that each time point and condition was studied with six biological replicates. Is this true for all data shown in any figure or table? I recommend adding the information on the number of replicates to each figure and table legend.
Ad 1) Yes, six plants were treated and such number of samples were taken for analysis. We added recommended information to relevant captions. Following sentence was added to the paragraph 2.1 (Results): “Rosette tissue samples were taken from six independent replicates of both treatments at six time points: 2, 4, 8, 16, 24, and 48 hours after the inoculation.”
2) Table 1 and table 2: For a better understanding the GO BPs of C24 and Col-0 should be edited and processes or genes that occur twice should be shortened, for example Table 2, items 6,7,and 8 are identical except for the description, this could be shortened. Table 1, item 8 (innate immune response) is a subgroup of item 9 (Immune response), only show one.
Ad 2) Following your request, in Table 2 items 7 and 8, as well as redundant items 16 and 17, were removed and the table was shortened. As for the Table 1, we would like to retain both entries (8 and 9) in the table, as they represent related GO terms with different number of members.
3) It in not fully clear from the table descriptions how the values were calculated. Was it Pst vs mock treatment at timepoint zero? Please add this information
Ad 3) At the beginning of the section 2.2. we have described the method for basal expression difference. Briefly, gene expression values in mock-treated Col-0 and C24 plants of the six time points were averaged, log2-transformed, and subtracted (Col-0 - C24) to calculate log-transformed ratios. Next (what is described in the “Methods” section), R-package GOstats was used to find the GO BP which were over-enriched. The cut-off p-value for significance level was considered as 0.05 after the FDR correction.
4) Section 2.3: Are there any datasets available? Maybe it was overlooked to add these to the manuscript or supplements, at least I could not find a link to any results.
Ad 4) An Excel table with the DEG’s comparison of the main (soil) and axenic experiment described in the section 2.3 was added as the supplementary table 5.
5) Figure 1B: The names are difficult to read, maybe without "unknown" metabolites or another arrangement of the letters these would be better accessible. What is the meaning of “107 variables” ?
Ad 5) The “107 variables” is the plotting software’s (R-package: EnhancedVulcano) feature informing how many analytes (variables) were displayed. For the ease of reading, analyte names were shortened (dropped derivatization description) “Unknown” in the description of an analyte was abbreviated to “Ukn”. The software attempts to distribute labels in a most unobtrusive way, hence some analyte labels were moved between the current and previous version of the figure.
6) Table 3: Showed a very long list which not easy to read. Maybe formatting was lost.
Ad 6) Following your recommendation, the Table 3 was shortened to present only glucosinolates displaying statistically significant changes 48h post infection or if a substance was detected in only one of the accessions. Other, for ease of finding, were highlighted in the Supplementary table 3.
7) Figure 3: Here in many cases the mock samples follow the Pst treated samples and expression increases. This is not clear to me, please explain. Figure legend: “Gene expression of genes”, remove the first “Gene”.
Ad 7) Mock expression profile following the Pst over the time, suggest that the enhanced gene expression might not necessary for activation of glucosinolates synthesis in Arabidopsis. The regulation of the pathway might occur at the level of posttranscriptional regulation, eg, phosphorylation of biosynthetic enzymes or ubiquitination of their protein partners. The figure highlights the differences between accessions.
Fig. 3 caption corrected
8) Table 4: Please mention why there are some genes written in bold letters.
Ad 8) This was an artifact of our data analysis. Bold type font removed.
9) It would be nice to have a picture of the infected vs mock treated plants. Are there symptoms visible after 48h ?
Ad 9) Unfortunately, no photographic documentation of the experiment is available.
10) Data handling: Are the omics data available to the public, except for the supplemental exel files?
Ad 9) This publication and the supplementary files will provide readers processed data. Raw data can be obtained through a direct reasonable request send to the corresponding authors. Gene expression data from main microarray experiment are publicly available at NCBI/GEO: accession GSE90852.
Reviewer 2 Report
In the research article title entitled “Transcriptomic and Metabolomic Analysis of a Pseudomonas- 2 Resistant versus a Susceptible Arabidopsis Accession”, the authors have integrated a multi-omics approach, to study the differences between the two Arabidopsis accessions Col-0 and C24 about their resistance to Pseudomonas syringae pathogen — before and after infection. The experiments were well designed, and the study shows many exciting results, including genotype-specific resistance, and post-Pst disease. The obtained results could be a potential resource for carrying functional characterization of genes involved in SAR mechanisms towards Pst resistance. In my view, the manuscript is well written, and the results were well justified in the discussion part, however, I would like to recommend the authors recheck the whole text for possible grammatical errors, and change gene names and gene Ids to italics. Along with that, I would suggest the authors consider the suggestions below.
1. The authors can show the phenotype of the Col-0 and Col-24 accessions before and after Pst infection.
2. Supplementary figure 2, is not cited anywhere in the text, check and correct.
Author Response
Dear Reviewer,
Thank you for your review of the manuscript. Following changes were applied to our manuscript.
- The authors can show the phenotype of the Col-0 and Col-24 accessions before and after Pst infection.
Ad.1. Unfortunately, no photographic documentation of the experiment is available.
- Supplementary figure 2, is not cited anywhere in the text, check and correct.
Ad.2. A reference to the Figure S3 in line 205 was corrected to refer to the Supplementary figure 2.